# Structural role for DNA Ligase IV in promoting the fidelity of non-homologous end joining

Benjamin M. Stinson[1,2], Sean M. Carney[1], Johannes C. Walter [1,2] ✉ & Joseph J. Loparo [1] ✉

Nonhomologous end joining (NHEJ), the primary pathway of vertebrate DNA double-strand-break (DSB) repair, directly re-ligates broken DNA ends. Damaged DSB ends that cannot be immediately re-ligated are modified by NHEJ processing enzymes, including error-prone polymerases and nucleases, to enable ligation. However, DSB ends that are initially compatible for re-ligation are typically joined without end processing. As both ligation and end processing occur in the short-range (SR) synaptic complex that closely aligns DNA ends, it remains unclear how ligation of compatible ends is prioritized over end processing. In this study, we identify structural interactions of the NHEJ-specific DNA Ligase IV (Lig4) within the SR complex that prioritize ligation and promote NHEJ fidelity. Mutational analysis demonstrates that Lig4 must bind DNA ends to form the SR complex. Furthermore, single-molecule experiments show that a single Lig4 binds both DNA ends at the instant of SR synapsis. Thus, Lig4 is poised to ligate compatible ends upon initial formation of the SR complex before error-prone processing. Our results provide a molecular basis for the fidelity of NHEJ.

Faithful repair of DNA double-strand breaks (DSBs) is critical for genome stability and tumor suppression. Cells employ two major pathways to repair DSBs: homologous recombination (HR) and nonhomologous end joining (NHEJ)[1]. HR uses a homologous template to ensure accurate restoration of the DNA sequence. In contrast, NHEJ directly re-ligates broken DNA ends without a template and thus functions in all phases of the cell cycle. Broken DNA ends are often damaged, which renders them incompatible for re-ligation. To overcome this problem, NHEJ utilizes an array of end-processing enzymes, including polymerases and nucleases, to resolve incompatible ends and allow ligation[2]. Such end processing is potentially mutagenic, but accumulating evidence suggests that it is restricted to incompatible ends, and undamaged compatible ends (i.e., blunt or "sticky") are typically joined without processing[3–8]. Thus, ligation is prioritized over end processing, thereby minimizing errors.

To promote ligation, core NHEJ factors recognize and align DNA ends in a process termed synapsis. The ring-shaped Ku70/80 heterodimer (Ku) initiates NHEJ by encircling broken DNA ends[9]. Ku serves as a recruitment hub for other NHEJ factors, including the DNA-dependent protein kinase catalytic subunit (DNA-PKcs), XRCC4-like factor (XLF), paralog of XLF and XRCC4 (PAXX), and a complex of XRCC4 and DNA Ligase IV (Lig4), which ultimately joins the DNA ends[2]. Recent single-molecule and structural studies have described distinct synaptic states formed by these core factors[10–12]. DNA ends are initially held in a "long-range" synaptic complex (LR complex), in which the DNA ends are tethered but not directly juxtaposed[10]. Ku and DNA-PKcs are sufficient for LR complex formation, but other LR complexes additionally incorporating PAXX and/or XLF have been observed[11,13]. The LR-complex transitions into a "short-range" synaptic complex (SR complex), in which the DNA ends are closely aligned for ligation.

[1]Department of Biological Chemistry and Molecular Pharmacology, Blavatnik Institute, Harvard Medical School, Boston, MA 02115, USA. [2]Howard Hughes Medical Institute, Boston, MA 02115, USA. ✉e-mail: johannes_walter@hms.harvard.edu; joseph_loparo@hms.harvard.edu

Efficient SR complex formation requires DNA-PKcs catalytic activity, a single XLF homodimer, and the Lig4-XRCC4 complex[10,14]. Notably, catalytically inactive Lig4 supports SR complex assembly[10,15] and even promotes subsequent DSB repair by other factors[16], suggesting that Lig4 plays a critical yet ill-defined structural role in SR complex formation.

We previously described how DNA end processing is coordinated with synapsis[8]. Using *Xenopus laevis* egg extracts, which efficiently recapitulate many properties of NHEJ observed in cells, we demonstrated that the activity of end-processing factors is largely restricted to the SR complex. From this observation, we proposed that because Lig4 is required to form the SR complex, compatible ends are ligated before processing occurs. A critical but untested assumption of this model is that Lig4 binds DNA ends in a ligation-competent state during initial SR complex formation.

Here, we demonstrate how interactions between Lig4 and DNA are coordinated with large-scale transitions of the NHEJ synaptic complex to promote fidelity of NHEJ. Using *Xenopus laevis* egg extracts, we show that Lig4 directly binds both DNA ends to assemble the SR complex. Mutational analysis demonstrates that SR complex formation requires Lig4 DNA binding. Additionally, single-molecule Förster resonance energy transfer (smFRET) experiments that simultaneously monitor DNA end synapsis and Lig4-DNA binding

demonstrate that Lig4 transiently binds DNA ends prior to SR synapsis, and a single Lig4 binds both DNA ends at the onset of SR synapsis. Thus, Lig4 is poised to ligate compatible ends immediately upon formation of the SR complex, thereby minimizing errors arising from unnecessary end processing.

## Results

### Characterization of Lig4 DNA-binding mutants

To test whether the Lig4-DNA interaction contributes to synapsis, we generated mutations in the DNA-binding domain (DBD) of Lig4. We selected six basic residues in the DBD of the *X. laevis* Lig4 ortholog that are expected, based on human Lig4-DNA structures[17], to make electrostatic interactions with DNA: K33, K35, K37, K167, R168, and K169 (Fig. 1a). We expressed and purified Lig4-XRCC4 complexes with multiple charge swap mutations: Lig4^mDBD1 (K33E, K35E, K37E), Lig4^mDBD2 (K167E, R168E, and K169E) and Lig4^mDBD1+2 (all six basic→Glu mutations) (Supplementary Data Fig. 1a). In addition, we purified a Lig4-XRCC4 complex lacking the DBD (Lig4^ΔDBD, residues 1-243 removed; Supplementary data Fig. 1a). We assessed DNA binding by these variants using a filter binding assay. Varying concentrations of Lig4-XRCC4 variants were incubated with a ³²P-labeled ~1 kb circular DNA, and samples were passed sequentially through nitrocellulose and positively charged nylon membranes. In this way, Lig4-XRCC4-DNA

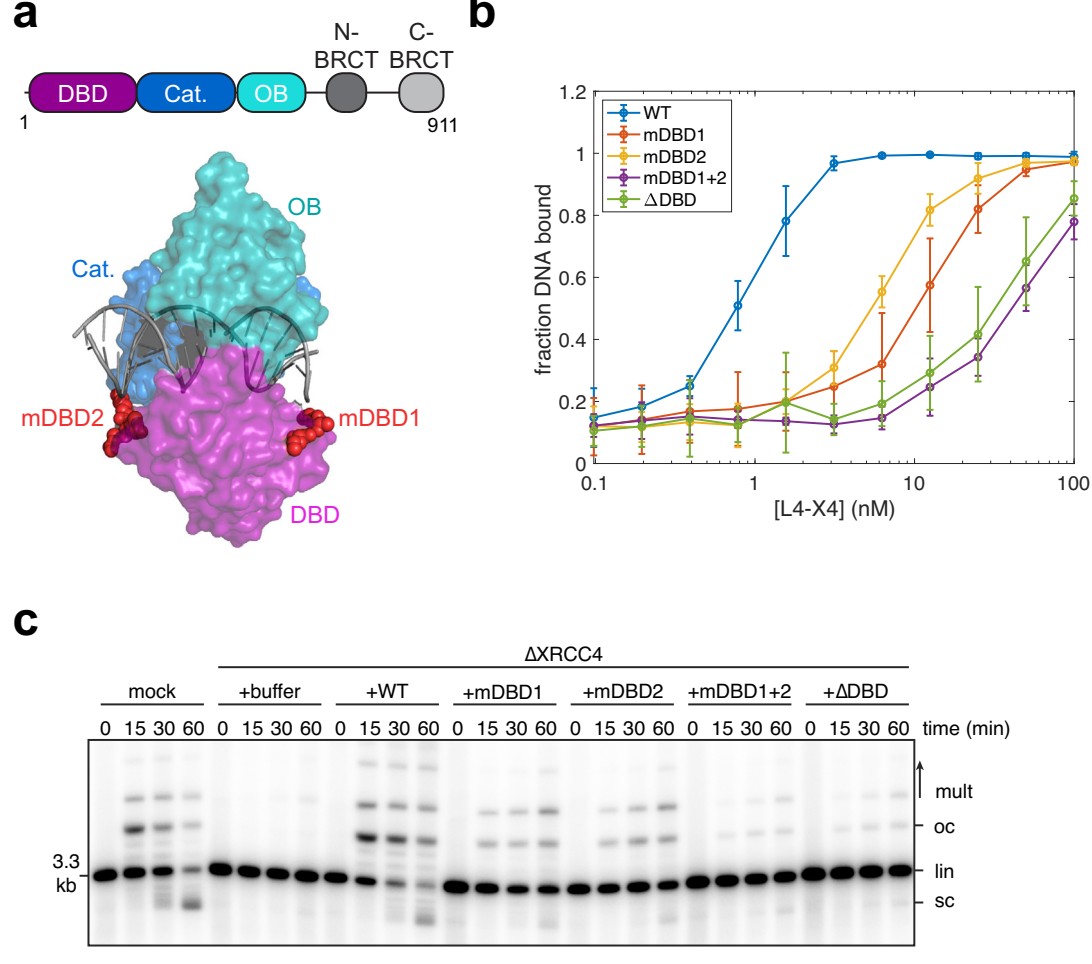

**Fig. 1 | Characterization of Lig4 DNA binding mutants. a** Domain structure of *Xenopus* Lig4 and X-ray crystal structure (PDB: 6BKG) of human Lig4. DBD: DNA-binding domain; Cat.: catalytic (adenylation) domain; OB: oligonucleotide/oligo-saccharide-binding fold domain. DBD point mutants identified in this study are highlighted in red. **b** Filter binding assay for DNA binding by Lig4 mutants, as detailed in Methods and Supplementary Data Fig. 1b. Data points represent the mean of three independent experiments; error bars, standard deviation. **c** Radiolabeled, blunt-ended, linear DNA molecules were added to the indicated extracts, and reaction samples were stopped at the indicated timepoints. Samples were analyzed by agarose gel electrophoresis and autoradiography. lin: linear; sc: supercoiled; oc: open circular; mult: multimers. Three independent experiments were performed and a representative autoradiogram is shown.

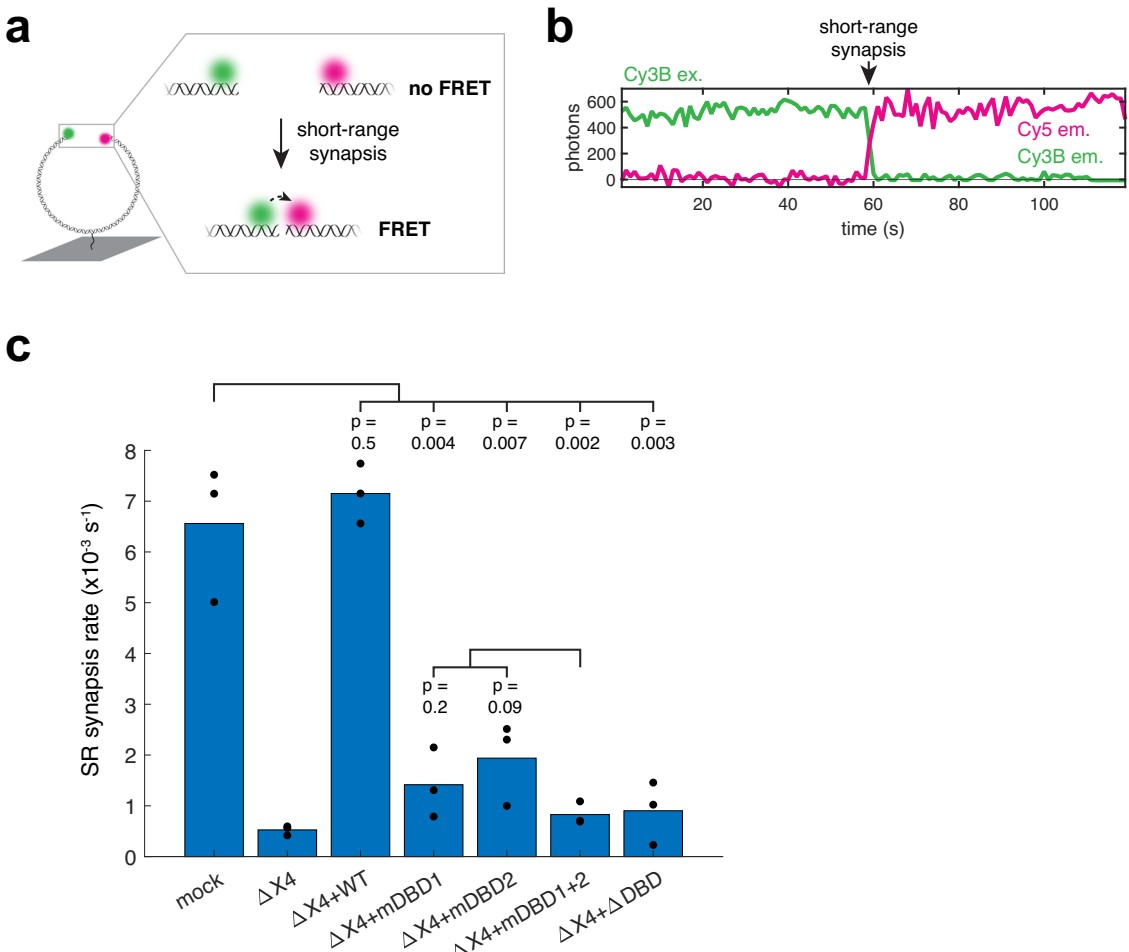

**Fig. 2 | DNA-binding by Lig4 is required for SR synapsis. a** Cartoon of smFRET assay for SR synapsis. Green circle: Cy3B fluorophore; magenta circle: Cy5 fluorophore; dotted arrow: energy transfer. **b** Representative single-molecule trajectory under Cy3B excitation depicting SR synapsis. Green line: Cy3B emission; magenta line: Cy5 emission. **c** Extracts were immunodepleted of Lig4-XRCC4 and supplemented with indicated recombinant Lig4-XRCC4 variants, and the rate of SR synapsis was measured using the assay depicted in panel **a**. Black dots: rates from three independent experiments; blue bars: average rates. *p*-values were calculated using the two-sided *t*-test.

complexes were first captured on the nitrocellulose membrane, and free DNA was captured on the nylon membrane to allow calculation of fractional DNA binding[18]. Lig4[mDBD1] and Lig4[mDBD2] bound DNA ~10-fold less strongly than wild-type Lig4, and Lig4[mDBD1+2] and Lig4[ΔDBD] bound DNA ~30-fold less strongly than wild-type (Fig. 1b, Supplementary Data Fig. 1b). Residual DNA binding even in the absence of the Lig4 DBD was likely mediated by the Lig4 catalytic and oligonucleotide/oligosaccharide-fold (OB) domains, both of which interact directly with DNA[17]. All Lig4 variants co-purified with XRCC4 (Supplementary Data Fig. 1a) and exhibited auto-adenylation rates within ~2-fold of that of wild-type Lig4 (Supplementary Data Fig. 1c), suggesting that the mutations introduced do not globally disrupt protein folding. Additionally, mass photometry experiments identified Lig4-XRCC4 complexes of the expected size with little aggregation (Supplementary Data Fig. 1d).

We then tested the ability of the Lig4 variants to support NHEJ. We immunodepleted egg extracts of endogenous Lig4-XRCC4 using an anti-XRCC4 antibody, which co-depletes Lig4[10]. Immunodepleted extracts were supplemented with recombinant Lig4-XRCC4 variants. Addition of radiolabeled linear, blunt-ended DNA followed by agarose electrophoresis and autoradiography allowed visualization of end joining. Wild-type Lig4-XRCC4 fully rescued end joining, and further addition of un-complexed XRCC4 did not stimulate end joining

(Supplementary Data Fig. 1e; compare lanes 9-12 and 17-20). NHEJ kinetics supported by the Lig4-XRCC4 DNA-binding mutants mirrored DNA binding affinity: Lig4[mDBD1] and Lig4[mDBD2] were modestly defective in end joining, and Lig4[mDBD1+2] and Lig4[ΔDBD] were severely defective (Fig. 1c). Together, these results establish that Lig4-XRCC4 DNA binding is essential for end joining.

## DNA-binding by Lig4 is required for SR synapsis

We next asked whether the Lig4-DNA interaction is required for SR complex assembly. To this end, we tested the Lig4 DNA-binding mutants in single-molecule Förster resonance energy transfer (smFRET) experiments that directly measure SR complex formation[10]. We labeled a ~3 kb blunt-ended DNA fragment with a Cy3B donor fluorophore near one end and a Cy5 acceptor fluorophore near the other (Fig. 2a). This DNA substrate was immobilized in a microfluidic flow cell on a streptavidin-functionalized coverslip via an internal biotin linkage. NHEJ was initiated by injecting egg extracts into the flow cell, and juxtaposition of DNA ends in the SR synaptic complex was monitored by FRET (Fig. 2b). Introduction of Cy3B and Cy5 fluorophores did not interfere with DNA ligation (Supplementary Data Fig. 2a). Additionally, high-FRET DNA molecules generally persisted following protein removal with 1% SDS, indicating that, although this assay cannot measure ligation in real time, DNA ends are efficiently

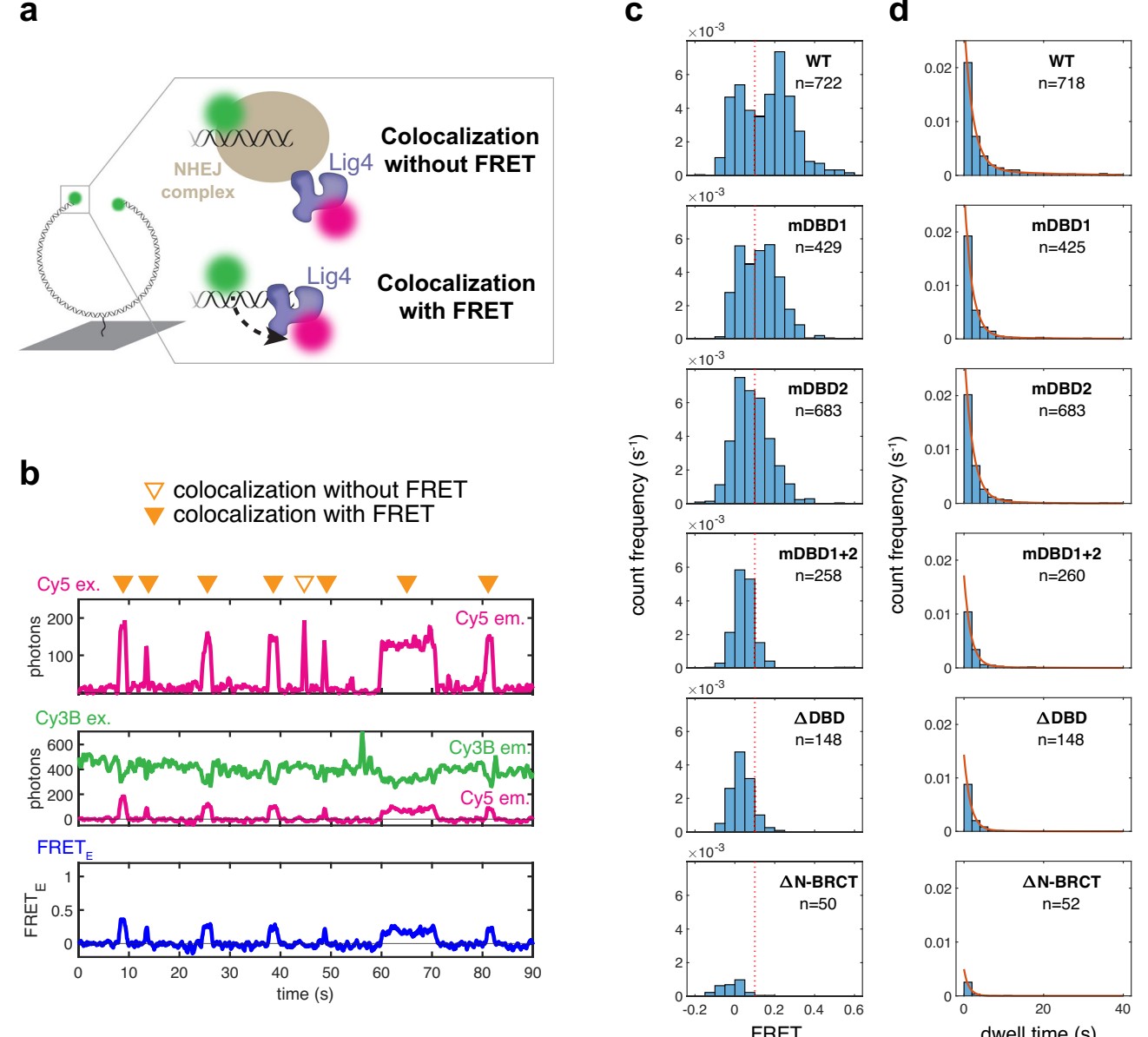

**Fig. 3 | DNA binding is required for Lig4 colocalization. a** Cartoon of smFRET assay for DNA end binding by Lig4. Green circles: Cy3B fluorophore; magenta circles: Cy5 fluorophore; dotted arrow: energy transfer. Colocalization occurs whenever Lig4 is recruited to the NHEJ complex, but FRET occurs only when Lig4 directly binds DNA ends. **b** Representative single-molecule trajectories showing Lig4 colocalization and end binding. Top panel: Cy5 excitation with Cy5 emission in magenta, measuring Lig4 colocalization; middle panel: Cy3B excitation with Cy3B emission in green and Cy5 emission in magenta, measuring Lig4 DNA binding; bottom panel: calculated FRET efficiency from middle panel. Orange arrows: colocalization events with or without FRET. **c** Colocalization events for each Lig4 variant were detected as described in Methods, and the average $FRET_E$ for each colocalization event was calculated and plotted on histograms for each Lig4 variant. Bin counts were normalized by total observation time, such that the y-axis reflects frequency of colocalization events. Data from three independent experiments for each Lig4 variant. **d** Histograms showing dwell time distributions for Lig4 variant colocalization events, detected as described in Methods. Bin counts were normalized by total observation time, such that the y-axis reflects frequency of colocalization events. Red lines show dwell time distribution fits generated by maximum likelihood estimation in MATLAB using one- or two-term exponential models, with fit parameters noted in Table 1. Data from three independent experiments for each Lig4 variant.

ligated in the high-FRET SR complex (Supplementary Data Fig. 2b). As reported previously[10], extracts depleted of Lig4-XRCC4 showed severely inhibited SR synapsis relative to mock-depleted extracts (Fig. 2c). Whereas addition of recombinant wild-type Lig4 fully rescued SR synapsis, addition of each DNA-binding defective mutant showed strongly reduced SR synapsis (Fig. 2c). Lig4[mDBD1] and Lig4[mDBD2] appeared to rescue synapsis to a slightly greater extent than Lig4[mDBD1+2], although this difference did not rise to the level of statistical significance (Fig. 2c). These results demonstrate that the Lig4-DNA interaction is critical for SR complex formation.

## DNA binding is required for Lig4 colocalization
To probe the dynamics of Lig4 recruitment and end binding during repair, we designed a smFRET assay to monitor the Lig4-DNA interaction directly (Fig. 3a). We modified the DNA substrate shown in Fig. 2a to include a Cy3B fluorophore at each DNA end, and we

**Table 1 | Lig4 colocalization dwell times**

| Lig4 variant | $\tau_1$ (s) [95% CI] | $\tau_2$ (s) [95% CI] | amp$_1$ (%) [95% CI] | amp$_2$ (%) [95% CI] |
|---|---|---|---|---|
| WT | 2.2 [1.9 2.5] | 15 [12 19] | 75 [68 82] | 25 [18 32] |
| mDBD1 | 1.9 [1.7 2.2] | 15 [9 21] | 89 [83 95] | 11 [5 17] |
| mDBD2 | 2.1 [1.8 2.4] | 10 [6 15] | 90 [83 97] | 10 [3 17] |
| mDBD1 + 2 | 1.7 [1.4 2.0] | 11 [5 18] | 91 [83 98] | 9 [2 17] |
| ΔDBD | 1.6 [1.3 2.0] | 13 [2 24] | 93 [85 100] | 7 [0 15] |
| ΔN-BRCT | 1.2 [0.9 1.6] | n.a. | 100 | n.a |

Fit parameters for dwell time distributions in Fig. 3d. Figures in brackets indicate 95% confidence intervals. All distributions were fit to two-term exponential models, except ΔN-BRCT dwells, which were fit to a one-term exponential model. τ, exponential time constant; amp., amplitude.

attached a Cy5 fluorophore to an N-terminal yBBR tag on Lig4 (Fig. 3a)[19]. Based on atomic-resolution structures of Lig4 bound to DNA[12,17], we expected that end binding by Lig4 would result in FRET from Cy3B to Cy5, whereas recruitment to the overall NHEJ complex without direct DNA binding would result in Cy5 colocalization without FRET (Fig. 3a). Extracts were depleted of endogenous Lig4-XRCC4, supplemented with wild-type Cy5-labeled Lig4-XRCC4, and introduced to the flow cell containing immobilized Cy3B-labeled DNA. An example single-molecule trajectory is shown in Fig. 3b, in which the top panel shows Lig4 colocalization with the substrate (Cy5 ex. / Cy5 em.), and the middle and bottom panels show Lig4-DNA binding (Cy3B ex. / Cy3B and Cy5 em. (middle); calculated FRET$_E$ (bottom)). Wild-type Cy5-labeled Lig4-XRCC4, which supported efficient end joining (Supplementary Data Fig. 2c), showed robust colocalization to DNA spots and minimal binding to coverslip locations that lacked a DNA signal, indicating that the observed colocalization events were highly specific (Supplementary Data Fig. 2d). Moving Cy3B to a position far from DNA ends did not affect Lig4 dwell times, indicating that the presence of Cy3B near DNA ends does not alter Lig4 binding (Supplementary Data Fig. 2e). ~70% of colocalization events exhibited Cy3B/Cy5 FRET (FRET$_E$ > 0.1), suggesting that wild-type Lig4 directly binds DNA ends for most colocalization events (Fig. 3c, WT). To verify that FRET resulted from Lig4-DNA binding, we repeated this experiment with Lig4 DNA-binding mutants. In all cases, perturbing the Lig4-DNA interaction resulted in a lower proportion of high-FRET colocalization events, with near total loss of the high-FRET population for the mDBD1 + 2 and ΔDBD mutants, as well as lower overall colocalization frequency (Fig. 3c, Supplementary Data Fig. 3). Colocalization frequency and the proportion of high-FRET events followed a similar trend as DNA binding, end joining, and rate of SR synapsis (Fig. 1b, c, and 2c; WT > mDBD1 ≈ mBDB2 > mDBD1 + 2 ≈ ΔDBD). These results validate Cy3B/Cy5 FRET in this assay as a reporter of Lig4-DNA binding, and they provide direct evidence that DNA binding is critical for Lig4-XRCC4 colocalization with the NHEJ complex.

To address whether Lig4 binds DNA ends before or after SR synapsis, we treated extracts with the DNA-PKcs inhibitor NU7441, which blocks formation of the SR complex[10]. Strikingly, Lig4-XRCC4 colocalization and end binding were essentially unaltered relative to the DMSO control (Supplementary Data Fig. 2f). Therefore, the vast majority of Lig4-DNA binding events observed occurred independently of SR synapsis. Taken together, our results suggest that Lig4 frequently binds DNA ends prior to SR synapsis. Therefore, the vast majority of Lig4 binding events do not result in successful ligation.

We next assessed the role of DNA binding in Lig4 colocalization dwell times. For wild-type Lig4, dwell times were biphasic, with a fast phase ($\tau_1$ ~ 2 s) accounting for ~75% of the distribution and a slower phase ($\tau_2$ ~ 15 s) accounting for the remainder (Fig. 3d, Table 1; Supplementary Data Fig. 2g shows single-phase fit). The presence of more than one phase likely reflects distinct populations of Lig4-XRCC4 that make different interactions with other NHEJ factors (see

Discussion). Dwell times were similar when Cy5 illumination power was increased (Supplementary Data Fig. 2h), suggesting that photo-bleaching did not substantially contribute to the observed kinetics. Absence of a 5′ phosphate on DNA ends did not alter dwell times, suggesting this moiety is not required for most Lig4 binding events (Supplementary Data Fig. 2i), although it may be important for productive synapsis[20]. Disruption of Lig4-DNA binding resulted in less frequent Lig4 recruitment with fewer long dwell times (Fig. 3d, amp$_2$ in Table 1). These results suggest that DNA binding is critical for stable Lig4-XRCC4 retention in the NHEJ complex. Consistent with this idea, wild-type Lig4 colocalization events with high FRET$_E$ (>0.1) were longer-lived than low FRET$_E$ colocalization events (Supplementary Data Fig. 2j). Overall, these results show that, despite the presence of multiple protein-protein interactions that are thought to recruit Lig4-XRCC4 to the NHEJ complex (see Discussion), Lig4 colocalization with the NHEJ complex is transient and dependent on DNA binding.

A region in the N-BRCT repeat of Lig4 interacts with Ku and is required for Lig4 recruitment[21]. Consistent with this, deletion of this region (residues 648-752) abrogated Lig4-XRCC4 colocalization with DNA ends (Fig. 3c, d), end synapsis (Supplementary Data Fig. 4c), and end joining (Supplementary Data Fig. 4b), even though DNA binding and adenylation activity were retained in this variant (Supplementary Data Fig. 1b, Supplementary Data Fig. 4a). Thus, both the Ku and DNA interactions of Lig4 are required for Lig4 colocalization, and neither alone is sufficient.

**A single Lig4 binds both DNA ends at the moment of SR synapsis**
Ligation requires that a single Lig4 binds both DNA ends. We wanted to determine whether such a structure is formed at the moment of SR synapsis, since this would promote rapid ligation of compatible ends before any end processing could occur. For this purpose, we developed a three-color single-molecule FRET assay that simultaneously monitors synapsis as well as Lig4 recruitment, stoichiometry, and DNA end binding. As in Fig. 2a, we labeled a blunt-ended DNA substrate with Cy3B and Cy5. Extracts were immunodepleted of endogenous Lig4-XRCC4, supplemented with Cy7-labeled Lig4-XRCC4, and introduced to flow cells containing immobilized Cy3B/Cy5 DNA substrate. Using alternating excitation of each dye, this assay monitors Lig4-XRCC4 recruitment (direct Cy7 excitation), SR synapsis (Cy3B→Cy5 FRET), and Lig4 binding of each DNA end (Cy3B→Cy7 FRET and Cy5→Cy7 FRET) (Fig. 4a). Lig4-XRCC4 stoichiometry was estimated by normalizing the Cy7 signal to the average stepwise change in Cy7 signal associated with a binding event (see Methods for details). The example trajectory in Fig. 4b shows a DNA molecule that does not undergo SR synapsis (low Cy3B→Cy5 FRET, top panel) but displays the following sequence of Lig4-DNA binding events (orange markers): (1) One Lig4-XRCC4 complex colocalizes (increase in Cy7 signal, bottom panel) and binds the Cy5 DNA end (Cy5→Cy7 FRET, middle panel); (2) Lig4-XRCC4 dissociates; (3) One Lig4-XRCC4 complex colocalizes and binds the Cy5 DNA end; (4) a second Lig4-XRCC4 complex colocalizes (increase in Cy7 signal, bottom panel) and binds the Cy3B DNA end (Cy3B→Cy7 FRET, top panel); (5) Lig4-XRCC4 bound to the Cy5 DNA end dissociates; (6) Lig4-XRCC4 bound to the Cy3B DNA end dissociates. As shown in this example, we frequently observed Lig4-XRCC4 DNA end binding (Cy3B→Cy7 FRET and/or Cy5→Cy7 FRET) prior to SR synapsis (low Cy3B→Cy5 FRET) (Fig. 4c and Supplementary Data Fig. 5a, orange boxes), consistent with the ability of Lig4 to bind DNA ends when SR synapsis is blocked (Supplementary Data Fig. 2c). Across all trajectories prior to SR synapsis, neither end was bound by Lig4 in ~60% of frames (neither Cy3B→Cy7 FRET nor Cy5→Cy7 FRET > 0.25; Fig. 4b markers 2 and 6; Fig. 4d, red box); one end bound in ~30% of frames (either Cy3B→Cy7 FRET or Cy5→Cy7 FRET; Fig. 4b, markers 1, 3, and 5; Fig. 4d, yellow boxes); or both ends bound in ~10% of frames (both Cy3B→Cy7 FRET and Cy5→Cy7 FRET; Fig. 4b, marker 4; Fig. 4d, green

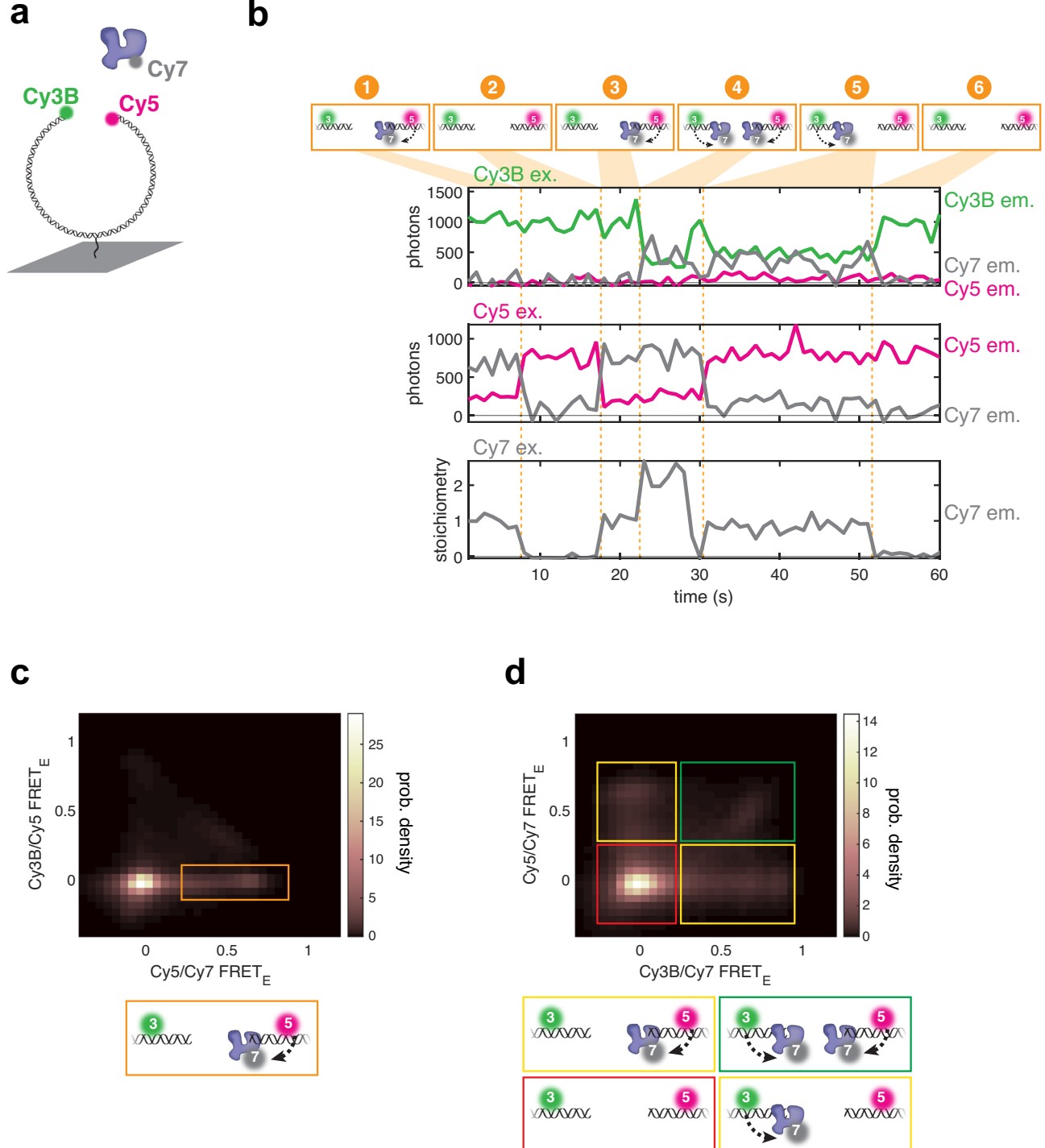

**Fig. 4 | Lig4 binds DNA ends outside the SR complex. a** Cartoon of three-color smFRET assay for synapsis and DNA end binding by Lig4. Green circle: Cy3B fluorphore; magenta circle: Cy5 fluorophore; gray circle: Cy7 fluorophore. **b** Representative single-molecule trajectories showing Lig4 colocalization and end binding. Top panel: Cy3B excitation with Cy3B emission in green, Cy5 emission in magenta, and Cy7 emission in gray; middle panel: Cy5 excitation with Cy5 emission in magenta and Cy7 emission in gray; bottom panel: Cy7 excitation with Cy7 stoichiometry in gray (see Methods for conversion from Cy7 emission to stoichiometry). See text for description of orange markers; energy transfer indicated by dotted arrows. **c** Heatmaps showing Cy3/Cy5 FRET as a function of Cy3B/

Cy7 FRET. Orange box shows Lig4 DNA binding prior to SR synapsis. Heatmaps contain data from all experimental frames in which neither Cy3B nor Cy5 had photobleached. Also see Supplementary Data Fig. 5a. **d** Heatmap showing Cy5/Cy7 FRET as a function of Cy3B/Cy7 FRET. Heatmap contains data from all experimental frames in which neither Cy3B nor Cy5 had photobleached and Cy3B/Cy5 FRET$_E$ was <0.15. Red box: neither Cy3B/Cy7 nor Cy5/Cy7 FRET, indicating neither end is bound by Lig4; yellow boxes: either Cy3B/Cy7 or Cy5/Cy7 FRET, indicating one end is bound by Lig4; green box: both Cy3B/Cy7 and Cy5/Cy7 FRET indicating both ends are bound by a separate Lig4.

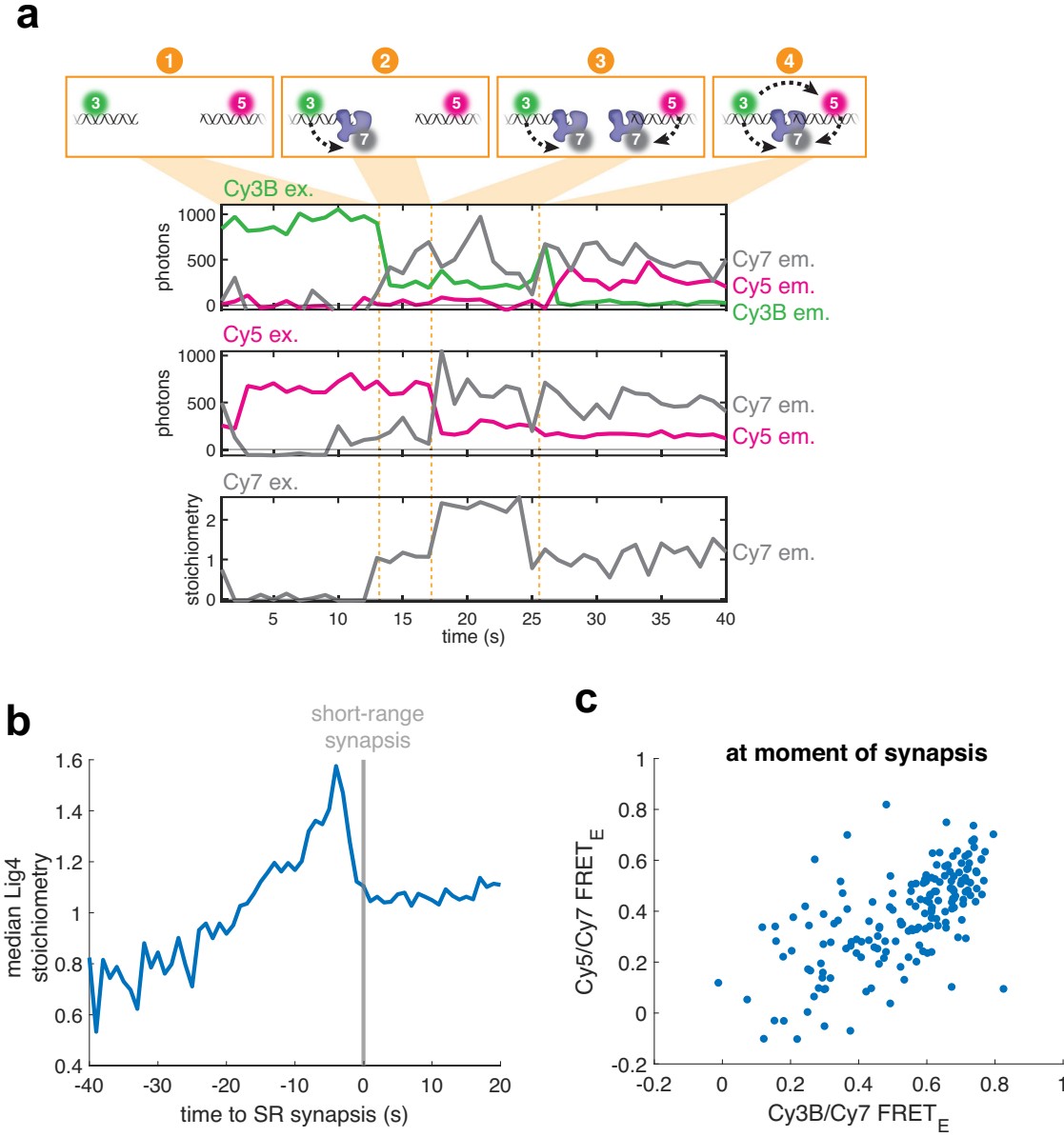

**Fig. 5 | A single Lig4 binds both DNA ends at the moment of SR synapsis.**
**a** Representative single-molecule trajectories showing Lig4 colocalization, Lig4 end binding, and synapsis. Panels as in Fig. 4b. **b** For molecules that underwent SR synapsis, Cy7 colocalization trajectories (e.g., bottom of panel **a**; $n = 248$ from 8 independent experiments) were aligned, with the onset of SR synapsis corresponding to $t = 0$ (gray bar). Blue line shows median Lig4 stoichiometry as a function of time to synapsis. See Supplementary Data Fig. 5b for distributions. **c** Scatter plot depicting Cy3B/Cy7 FRET and Cy5/Cy7 FRET at the moment of SR synapsis, as detected by a stepwise increase in Cy3B/Cy5 FRET using the MATLAB ischange function. Only synapsis events for which colocalization of Cy7-labeled Lig4 was detected are shown. Likely due to incomplete Cy7 labeling, Lig4 colocalization was detected in 178 of 248 total synapsis events. $n = 178$ from 8 independent experiments.

box). Thus, prior to SR synapsis, DNA molecules usually have at most one end bound by Lig4-XRCC4, although occasionally two Lig4-XRCC4 complexes are recruited, with each engaging one end.

Although recruitment of two L4-XRCC4 complexes was rare overall, we frequently observed arrival of two Lig4-XRCC4 complexes shortly before SR synapsis (as in Fig. 5a, marker 3), with one of the two complexes dissociating immediately prior to or concurrent with synapsis (as in Fig. 5a, marker 4; top panel shows Cy3B→Cy5 FRET indicating SR complex formation). The time resolution of this experiment did not permit identification of which Lig4 molecule dissociated before the remaining Lig4 mediated SR synapsis. To visualize Lig4-XRCC4 stoichiometry across all observed synapsis events, we aligned trajectories at the instant of SR synapsis (Fig. 5b, gray line) and plotted

the median Lig4-XRCC4 stoichiometry (Fig. 5b; stoichiometry distributions are shown in Supplementary Data Fig. 5b). 5–10 s prior to SR synapsis, Lig4-XRCC4 was present at a stoichiometry of ~1.6 before dropping to a stoichiometry of ~1 at the instant of synapsis. Thus, although the NHEJ complex frequently contained two Lig4-XRCC4 complexes before SR synapsis, one of these complexes dissociated to leave a single Lig4-XRCC4 at the onset of synapsis. Consistent with this idea, we detected a Lig4-XRCC4 dissociation event within the 10 s period before synapsis for ~40% of synapsis events (Supplementary Data Fig. 5c; example in Fig. 5a, markers 3-4), an underestimate due to incomplete Cy7 labeling (~70%; Supplementary Data Fig. 5d shows an example trajectory in which no Lig4-XRCC4 dissociation was detected prior to synapsis). Our results are consistent with the transient

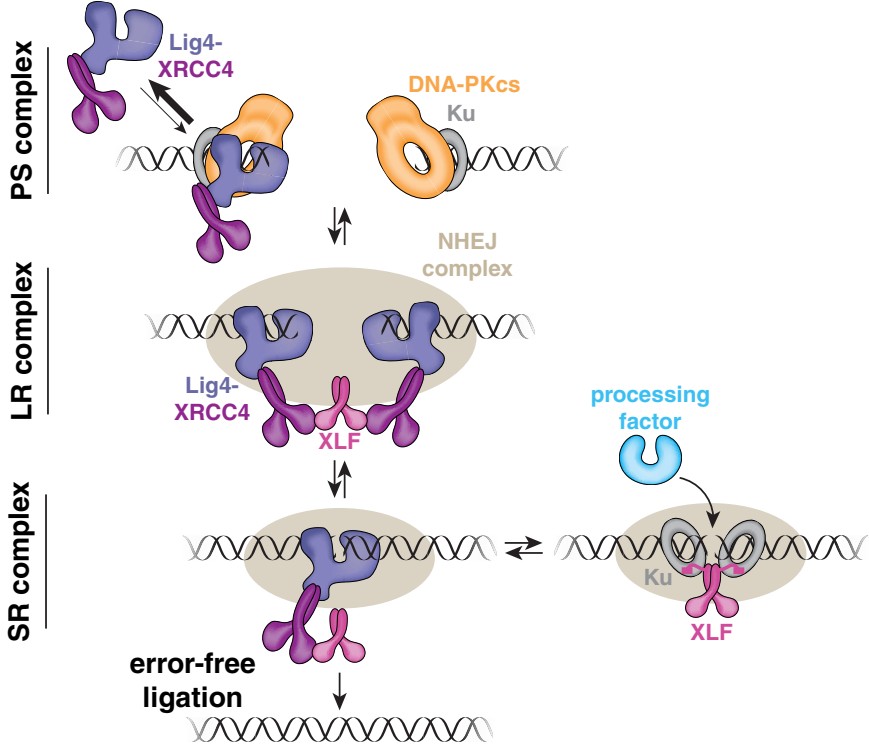

**Fig. 6 | Model of Lig4-DNA interaction during synapsis.** In the presynaptic (PS) complex and/or long-range (LR) synaptic complex, Lig4 transiently binds DNA ends. We postulate that although DNA-PKcs usually occludes DNA ends (top right), DNA ends are transiently accessible and captured by Lig4 (top left), which is enriched near DNA ends due to interactions with Ku and potentially other NHEJ factors (see text). 5-10 s prior to SR synapsis, both DNA ends are frequently bound by distinct Lig4-XRCC4 complexes, perhaps allowing formation of an XRCC4-XLF-XRCC4 bridge (see text). Immediately before or concurrent with SR synapsis, one of the Lig4-XRCC4 complexes dissociates such that a single Lig4 binds both DNA ends at the instant of SR synapsis. Compatible ends can be rapidly joined without errors from this ligation-competent state. If ends are incompatible, Lig4 must dissociate to allow end processing, and XLF may interact with Ku on both sides of the DSB to maintain SR synapsis. This may involve interactions between Ku and the XLF Ku-binding-motif on the XLF C-terminal tail (pink box at end of curved line) and/or interactions between Ku and the XLF coiled-coil stalk[12].

formation of an XRCC4-XLF-XRCC4 "bridge" (see Discussion) before a single Lig4 mediates SR synapsis.

Finally, we analyzed the Lig4-DNA interaction at the instant of SR synapsis. Most synapsis events were accompanied by high FRET efficiency in both the Cy3B→Cy7 and Cy5→Cy7 channels (Fig. 5c; example in Fig. 5a, marker 4). These results suggest that the SR complex is ligation-competent at the moment it forms, as a single Lig4-XRCC4 complex directly binds both DNA ends.

## Discussion

We and others showed previously that Lig4-XRCC4 plays a critical, non-catalytic function during synapsis[10,11,15,16,22,23]. Here, we elucidate this structural role by directly visualizing the Lig4-XRCC4 interaction with DNA in real-time. Our results support a model in which SR synapsis is tightly coupled to productive interactions between Lig4 and DNA ends that render NHEJ error-resistant (Fig. 6). In the presynaptic and/or long-range synaptic complex, Lig4 transiently binds DNA ends, and the Lig4-DNA interaction is critical for retention of Lig4-XRCC4 in the NHEJ complex. Within the long-range synaptic complex, shortly before the transition to the SR complex, DNA ends are bound by two separate Lig4-XRCC4 complexes, consistent with the formation of an XRCC4-XLF-XRCC4 bridge that is critical for SR synapsis. Subsequently, a single Lig4 binds to both DNA ends to drive formation of the SR complex. This moment represents a critical opportunity for error-free ligation of compatible ends, because DNA ends pass through an intermediate that has a high probability of ligation before the onset of end processing, which is restricted to the SR complex[8]. Overall, our results provide a mechanistic basis for the high fidelity of NHEJ, which is observed in multiple model systems[3–8].

Lig4-DNA binding prior to SR synapsis was unexpected, as blunt DNA ends are occluded by DNA-PKcs in X-ray and cryo-EM structures of early-stage NHEJ complexes[12,13,24–27]. DNA-PKcs autophosphorylation allows processing enzymes to access DNA ends[27–30], yet we observe robust Lig4-DNA binding even in the presence of DNA-PKcs inhibitor. We hypothesize that DNA ends are transiently accessible for Lig4 binding prior to DNA-PKcs autophosphorylation. This state may be difficult to resolve by structural methods due to its transient nature. Given that disruption of individual interactions (Lig4-DNA, Ku-Lig4[N-BRCT]) abrogates Lig4-XRCC4 recruitment to the NHEJ complex, Lig4-DNA binding prior to SR synapsis appears to require multiple, weak interactions. We propose that the Ku-Lig4[N-BRCT] interaction initially recruits Lig4 and allows it to capture transiently accessible DNA ends. Lig[DBD]-Ku[12] and XRCC4-XLF-XRCC4[31] interactions likely also contribute to Lig4 enrichment. In this way, multiple protein-protein interactions allow Lig4 unique access to DNA ends prior to SR synapsis, whereas processing enzymes lacking these interactions are excluded.

Our results suggest that transition to the SR synaptic complex requires transient formation of an XRCC4-XLF-XRCC4 bridge—a single XLF homodimer interacting with two Lig4-XRCC4 complexes (Fig. 6). Previously, we showed that XLF mutations that disrupt one of the two XRCC4 binding sites strongly attenuate SR synapsis, consistent with a critical role for an XRCC4-XLF-XRCC4 bridge[14]. Moreover, this bridge is observed in both long-range and SR complex cryo-EM structures[12,13]. Remarkably, and in agreement with the above evidence, we frequently observe two Lig4-XRCC4 complexes 5–10 s prior to SR synapsis. Notably, XLF is stably recruited to the NHEJ complex within a similar time interval as Lig4-XRCC4 prior to SR synapsis[14]. We hypothesize that this Lig4-XRCC4 stoichiometry represents the formation of the

XRCC4-XLF-XRCC4 bridge. We further speculate that the XRCC4-XLF-XRCC4 bridge serves multiple functions as an intermediate to the SR complex, including tethering the ends together, stabilizing XLF and/or Lig4-XRCC4, and/or properly positioning these factors for SR synapsis.

Although multiple lines of evidence point to a critical role of the XRCC4-XLF-XRCC4 bridge, our real-time single-molecule imaging shows that the bridge is transient and does not persist in the SR complex, as one of the Lig4-XRCC4 complexes dissociates near the moment of SR synapsis. Consistent with this finding, our prior work showed that although attenuation of the XLF-XRCC4 interaction impedes initial assembly of the SR complex, it does not affect SR complex stability[14]. We propose that dissolution of the XRCC4-XLF-XRCC4 bridge is critical to allow formation of the SR complex. In the bridge structure, both DNA ends are bound by separate Lig4-XRCC4 complexes (Fig. 6); however, one of these Lig4-XRCC4 complexes must dissociate to allow a single Lig4-XRCC4 to bind both DNA ends, which we show occurs in the SR complex. Thus, dynamic Lig4-DNA binding facilitates efficient SR synapsis. The multivalent network of weak interactions required for Lig4-DNA binding likely enables these important Lig4 dynamics, as such interaction networks allow specific yet reversible binding[32].

The presence of only a single Lig4-XRCC4 at the instant of SR synapsis contrasts with a cryo-EM structure of a SR complex, which includes two Lig4-XRCC4 complexes and an XRCC4-XLF-XRCC4 bridge[12]. However, consistent with our results, the structure shows only one Lig4 catalytic core binding both DNA ends; the second Lig4 does not appear to bind DNA, with only the BRCT domains resolved. Our results predict that in the absence of the Lig4-DNA interaction, the second Lig4-XRCC4 would rapidly dissociate and the XRCC4-XLF-XRCC4 bridge would disassemble. The kinase activity of DNA-PKcs, which is present in our system but was omitted from the cryo-EM reconstitution of the SR complex, may influence the stability of the XRCC4-XLF-XRCC4 bridge and therefore of Lig4, as phosphomimetic substitutions in XRCC4 and XLF attenuate DNA bridging by these proteins[33]. Whereas the cryo-EM study suggests that the presence of two Lig4-XRCC4 complexes is crucial to allow tandem ligation of both DNA strands, our observation that a single Lig4-XRCC4 is present at the moment of synapsis suggests that ligation of the two strands need not be tightly coupled. Consistent with this idea, studies using human cells, human cell extracts, and purified proteins report single-ligation NHEJ intermediates[34–36]. Thus, the most critical function of NHEJ may be rapid ligation of a single DNA strand, thereby re-establishing a covalent connection between broken DNA ends[7].

Our results raise new questions about how interactions among NHEJ factors evolve over the course of synapsis and repair. We demonstrate that the binding of Lig4 to both DNA ends drives formation of the SR complex—a state that is responsible for both end processing, and ultimately, ligation[8]. To allow access by processing enzymes, Lig4-XRCC4 likely must release the DNA ends, which would result in Lig4 dissociation from the NHEJ complex. In the absence of the Lig4-DNA interaction, however, it is unclear how SR synapsis is maintained. An intriguing possibility is that the XLF homodimer, which is retained following SR synapsis[14] and forms multiple contacts with Ku on either side of the DSB[12,37], fulfils this role (Fig. 6). Subsequent Lig4 re-binding would allow ligation of appropriately processed DNA ends. Further studies will be required to fully describe the dynamic intermediates underlying faithful NHEJ.

## Methods

### Egg extract preparation
High-speed supernatant (HSS) of egg cytosol was prepared as described[38].

### Preparation of DNA substrates for end joining
Radiolabeled DNA substrates were prepared were prepared in the following manner. pBMS6, a derivative of pBlueScript, was linearized

with Acc65I (New England Biolabs) and resulting 5′ overhangs were filled in using Klenow Fragment polymerase and a mixture of dNTPs including α-$^{32}$P-dATP. 20 μL fill-in reactions contained 1x NEBuffer 2.1 (New England Biolabs); 1 μg linearized plasmid; 33 μM each dTTP, dGTP and dATP; 1 μL α-$^{32}$P-dATP (Perkin Elmer; 3000 Ci/mmol, 10 mCi/ml); and 0.5 μl 5 U/μl Klenow (New England Biolabs). Reactions proceeded for 15 min at room temperature. The resulting radiolabeled, blunt-ended, linear DNA fragment was recovered using a Qiagen PCR clean-up kit.

Fluorescently labeled DNA substrates were prepared in the following manner. pBMS6, a derivative of pBlueScript, was linearized with SphI and AatII (New England Biolabs). The resulting 2977 bp fragment was separated on a 1x TBE agarose gel and recovered by electroelution and ethanol precipitation. Duplex oligonucleotide adapters (Integrated DNA Technologies, Inc.) with appropriate fluorophore modifications (see below) were ligated to each side of this backbone fragment to generate the described DNA substrates. To generate duplex adapters, oligonucleotide stocks (10 mM) in annealing buffer (10 mM Tris, pH 8.0, 50 mM NaCl, 1 mM EDTA) were combined in equal volumes and annealed by heating to 95 °C for 2 min then slowly cooling to room temperature in 1 °C steps lasting 60 s. One duplex adaptor (5′PO$_4$- CGTACCGC/iAmMC6T/CTAT annealed to 5′OH-ATA-GAGCGGTACGCATG) contained an overhang complementary to that generated by SphI; similarly, the second duplex adaptor (5′PO$_4$-GGCGGTAT/iAmMC6T/CACG annealed to 5′OH-CGTGAATACCGC CACGT) contained an overhang complementary to that generated by AatII. Duplex adapters (250 nM) were ligated to the backbone fragment (25 nM) with T4 DNA ligase (New England Biolabs). The desired ~3 kb product was separated on a 1x TBE agarose gel and recovered by electroelution and ethanol precipitation (fluorescent DNA substrates) or with the QIAquick gel extraction kit (QIAGEN).

Fluorescently labeled DNA substrates were prepared as previously described[8]. In brief, pBMS6 was linearized with SphI and AatII (New England Biolabs). The resulting 2977 bp fragment was separated on a 1x TBE agarose gel and recovered by electroelution and ethanol precipitation. Duplex oligonucleotide adapters (Integrated DNA Technologies, Inc.) were ligated to each side of this backbone fragment to generate blunt-ended DNA substrates and purified by agarose gel electrophoresis. (SphI-compatible duplex: CGTACCGC/iAmMC6T/ CTAT annealed to ATAGAGCGGTACGCATG; AatII-compatible duplex: GGCGGTAT/iAmMC6T/CACG annealed to CGTGAATACCGCCACGT). DNA was recovered by electroelution and ethanol precipitation and subsequently was treated with Nt.BbvCI (New England Biolabs) to introduce two nicks on the same stand near the middle of the molecule, thereby allowing removal of a 25-mer oligonucleotide. A 10-fold molar excess of an internally biotinylated, 5′-phosphorylated oligonucleotide with the same sequence was then added to the digestion mixture, annealed, and ligated into the gap. DNA substrates were purified by agarose gel electrophoresis and recovered by electroelution and ethanol precipitation. Finally, except for the 5′ OH DNA in Supplementary Data Fig. 2i, DNA substrates were phosphorylated with T4 PNK prior to use.

### Preparation of fluorescently-labeled oligonucleotides
Amino-modified synthetic oligonucleotides were reacted with NHS-ester fluorophore derivatives. A typical labeling reaction contained 46 mL labeling buffer (100 mM sodium tetraborate, pH 8.5), 2 mL amino-modified oligo (25 mg/mL stock in water), and 2 mL NHS-ester fluorophore (50 mg/mL stock in DMSO). The reaction was allowed to proceed overnight at room temperature in the dark. The mixture was ethanol precipitated to remove excess fluorophore. The pellet was resuspended in 10 mL Gel Loading Buffer II (Invitrogen) and subjected to denaturing PAGE electrophoresis on a 20% Urea-PAGE gel. The band corresponding to the labeled oligo was excised and crushed by centrifugation through a 1.7 mL microcentrifuge tube with a hole in the

bottom made with an 18-gauge needle. 500 mL TE buffer (10 mM Tris, pH 8.0, 0.1 mM EDTA) was added to the crushed gel slice, and the mixture was frozen in liquid nitrogen, rapidly thawed in warm water, and rotated overnight at room temperature in the dark. The solution was collected, and the labeled oligo was recovered by ethanol precipitation.

## Protein expression and purification

The following protocol was used for expression and purification of all L4-XRCC4 variants. Both sets of L4-XRCC4 variants (unlabeled and Cy5-labeled) were expressed and purified in parallel. pETDuet-1 plasmids were constructed encoding *Xenopus* Lig4 and XRCC4 and are available from the corresponding authors upon reasonable request. Lig4 contained a TEV-cleavable N-terminal His$_6$ tag and a 3 C protease cleavable C-terminal TwinStrep tag. XRCC4 was untagged. Expression plasmids were transformed into BL21 cells and plated on LB-agar plates containing 100 µg/mL ampicillin. Single colonies were used to inoculate 5 mL LB cultures supplemented with 100 µg/mL ampicillin, which were grown overnight at 37 °C. These starter cultures were added to 250 mL Terrific Broth supplemented with 100 µg/mL ampicillin, and cultures were shaken at 37 °C to an optical density of 1.2. Cultures were moved to a shaker at 16 °C, and 1 mM IPTG was added to induce expression overnight. Cultures were harvested by centrifugation and resuspended in ice-cold 20 mL lysis/wash buffer (20 mM Tris, pH 8,0; 400 mM NaCl; 10 mM imidazole; 1 mM DTT; 10% glycerol). A Roche cOmplete Protease Inhibitor Cocktail (EDTA free) tablet was added to the resuspension, and cells were lysed by sonication on ice. The lysate was centrifuged at 50,000 g for 60 min at 4 °C, and the supernatant was added to 0.5 mL (bed volume) Ni-NTA resin (Qiagen) equilibrated in wash/lysis buffer. The resulting mixture was rotated at 4 °C for 60 min and then poured into a 1 mL polypropylene column (Qiagen). The resin was washed three times with 5 mL lysis/wash buffer and eluted in 0.25 mL fractions with Ni-NTA elution buffer (composition same as lysis/wash buffer, but with 250 mM imidazole). Fractions were analyzed by SDS-PAGE, and fractions containing Lig4-XRCC4 were applied to a 0.5 mL (bed volume) Streptactin XT column (IBA Biosciences) equilibrated in lysis/wash buffer. The Streptactin XT column was washed three times with lysis/wash buffer and eluted in 0.5 mL fractions with 1x BXT buffer (IBA Biosciences). Fractions containing Lig4-XRCC4 were concentrated with a 10 kDa MWCO Amicon centrifugal filter unit and loaded onto a Superdex200 Increase 10/300 GL (Cytiva) column equilibrated in SEC buffer (20 mM HEPES, pH 7.5; 150 mM NaCl; 1 mM DTT; 10% glycerol). Fractions containing pure Lig4-XRCC4 were concentrated with a 10 kDa MWCO Amicon centrifugal filter unit, aliquoted, flash frozen in liquid nitrogen, and stored at −80 °C. Protein concentrations were estimated by absorbance at 280 nm.

For fluorescently-labeled Lig4, the 11 amino acid ybbR tag[19] was inserted between the TEV recognition site and Lig4 CDS, and labeling took place prior to gel filtration chromatography. Concentrated fractions from the Streptactin column were supplemented with final concentrations of 10% glycerol (v/v), 10 mM MgCl2, 5 µM Sfp synthase (plasmid obtained from Addgene (pET-Sfp, #159617) and purified as described[19]), and 100 µM CoA-Cy5 or CoA-Cy7 (prepared as described below). Labeling reactions were carried out in the dark overnight at 4 °C, and labeling reactions were loaded directly onto a Superdex200 Increase column as described above. Labeling efficiencies were 60-80% as estimated by absorbance at the dye absorbance maximum.

Un-complexed XRCC4 homodimer (Supplementary Data Fig. 2e) was a gift from T. Graham[14].

## Preparation of Coenzyme A-fluorophore conjugates

CoA-dye conjugates were prepared essentially as described[19], with a modified purification procedure. Briefly, 0.5 mg maleimide-functionalized sulfonated Cy5 or Cy7 (Lumiprobe cat. # 13320 or 15320) in 125 µL DMSO was added to 1 mg CoA trilithium salt in 375 µL 100 mM HEPES, pH 7.0. The reaction was allowed to proceed in the dark at room temperature for 1 hour. The reaction mixture was loaded onto two 1 mL HiTrapQ columns (Cytiva) connected in series and equilibrated in 10% acetonitrile (v/v). CoA conjugates were purified using a 0-20% gradient over 20 column volumes (A: 10% acetonitrile; B: 3 M LiCl), with desired conjugates eluting as the final peak of the chromatogram. Fractions from this peak was pooled and CoA conjugates were precipitated with 20 volumes cold acetone and collected by centrifugation. Pellets were washed with cold acetone and resuspended in 10 mM Tris, pH 8.0 CoA conjugate concentrations were estimated by absorbance at the dye absorbance maximum.

## Filter binding assay for DNA binding

Filter binding experiments used a radiolabeled 1 kb circular DNA substrate to preclude complications from ligation of the DNA substrate. As described above in "Preparation of DNA substrates for end joining," pBMS6 was linearized with Acc65I (New England Biolabs) and resulting 5′ overhangs were filled in using Klenow Fragment polymerase and a mixture of dNTPs including α-$^{32}$P-dATP. The resulting radiolabeled, blunt-ended, linear DNA fragment was recovered using a Qiagen PCR clean-up kit, diluted to ~1 nM in 1x T4 DNA ligase reaction buffer (New England Biolabs), and treated with T4 DNA ligase to circularize the DNA. Ligation reactions were concentrated using a 10 kDa MWCO Amicon centrifugal filter unit and separated on a 1% agarose gel containing ethidium bromide by electrophoresis. The supercoiled 1 kb band was excised and purified using a Qiagen gel extraction kit.

The filter binding assay protocol was modified from reference[18]. Hybond-N+ (Cytiva) and nitrocellulose (Whatman) membranes were equilibrated in a 1:3 mixture of SEC buffer:FB buffer (50 mM Tris, pH 8.0; 1 mM MgCl$_2$). Membranes were assembled in a Biorad Bio-Dot apparatus with the nitrocellulose membrane on top of the Hybond membrane. Serial 2-fold dilutions of Lig4-XRCC4 variants were prepared in SEC buffer in a multi-well plate, and the DNA substrate was diluted to 6 pg/mL in FB buffer. Binding reactions were initiated by adding 15 µL of the DNA mixture to 5 µL of the protein mixture and were allowed to proceed for 5 min at room temperature. Reactions were transferred to the Bio-Dot apparatus with a multi-channel pipette, and vacuum was immediately applied to draw samples through the membranes. Each well was washed twice by adding 100 µL ice-cold 1:3 SEC buffer:FB buffer and rapidly applying vacuum. Membranes were dried on a vacuum gel drier and exposed to storage phosphor screens for autoradiography and imaged with a Typhoon FLA 7000 imager (GE Healthcare).

## Lig4 autoadenylation assay

Autoadenylation assays were performed as described previously[10] with slight modifications. In a 10 µL reaction volume, 120 nM Cy5-labeled Lig4-XRCC4 variants were treated with 50 µM inorganic pyrophosphate in adenylation buffer (60 mM Tris, pH 8.0; 10 mM MgCl$_2$; 5 mM DTT; 5 µg/mL BSA; 10% glycerol) for 2 min at room temperature to deadenylate Lig4. 0.5 µL α-$^{32}$P-ATP (3000 Ci/mmol, 10 mCi/ml; Perkin Elmer) was added to each sample. Adenylation was allowed to proceed at room temperature and quenched at indicated timepoints with one volume of 2x Laemmli sample buffer. Samples were separated by SDS-PAGE, and gels were fixed with 10% methanol/10% acetic acid before being dried on a vacuum gel drier, exposed to storage phosphor screens for autoradiography. Storage phosphor screens were imaged with a Typhoon FLA 7000 phosphorimager (GE Healthcare), and Cy5 signal was measured with an AI600 Imager (GE Healthcare). To account for variable recovery of Lig4, autoradiogram signal was normalized by the Cy5 signal. Relative adenylation was then normalized to the level of wild-type Lig4 adenylation at 10 min.

## Mass photometry

Mass photometry experiments were performed using the Refeyn TwoMP instrument. To improve particle adsorption, coverslips were placed in a coplin jar, immersed in 1 M KOH, and placed in an ultrasonic water bath for 60 min. To generate a standard mass curve, thyroglobulin and beta-amylase were diluted to 3 nM and 10 nM, respectively, in SEC buffer. Thyroglobulin, beta-amylase dimer, and beta-amylase tetramer peaks were selected as calibrants. Lig4-XRCC4 complexes were diluted to 10 nM in SEC buffer and measurements were collected for 20 sec.

## Antibodies and immunodepletion

The rabbit polyclonal antibody raised against XRCC4 was previously described and is available from the corresponding authors upon reasonable request[10]. XRCC4 immunodepletions were carried out using the following protocol: 3 volumes of 1 mg/mL affinity-purified antibody was gently rotated with 1 volume Protein A Sepharose beads (GE Healthcare) overnight at 4 °C or 1 hour at room temperature. Beads were washed extensively with ELBS (2.5 mM MgCl2, 50 mM KCl, 10 mM HEPES, pH 7.7, 0.25 M sucrose), and ten volumes of egg extract containing 7.5 ng/μL nocodazole were immunodepleted by two rounds of gentle rotation with one volume of antibody-bound beads for 20 min at room temperature. Immunodepleted extracts were either used immediately or aliquoted and flash-frozen in liquid nitrogen.

## Ensemble NHEJ assays

Ensemble NHEJ assays were conducted at room temperature. Egg extracts were supplemented with the following (final concentration indicated in parentheses): pBMS6 (30 ng/μL); ATP (3 mM); phosphocreatine (15 mM); and creatine phosphokinase (0.01 mg/mL; Sigma). Joining reactions were initiated by addition of 2.5 ng/μL radiolabeled linear DNA substrate (final concentration, prepared as described above).

For analysis by agarose gel electrophoresis, samples were withdrawn at the indicated times and mixed with a 2.5 volumes agarose stop solution (80 mM Tris, pH 8.0, 8 mM EDTA, 0.13% phosphoric acid, 10% Ficoll, 5% SDS, 0.2% bromophenol blue). Samples were treated with Proteinase K (1.4 mg/mL final concentration) for 60 min at 37 °C or room temperature overnight, and products were separated by electrophoresis on a 1x TBE 0.8% agarose gel. Gels were dried under vacuum on a HyBond N+ membrane (GE Healthcare) and exposed to a storage phosphor screen, which was imaged with a Typhoon FLA 7000 imager (GE Healthcare).

## Single-molecule microscope, chamber preparation, and general protocol

Samples were imaged with a through-objective TIRF microscope built around an Olympus IX-71 inverted microscope body[39]. 532 nm, 641 nm, and 730 nm laser beams (Coherent Sapphire 532, Cube 641, and OBIS 730, respectively) were expanded, combined with dichroic mirrors, expanded again, and focused on the rear focal plane of an oil-immersion objective (Olympus UPlanSApo, 100×; NA, 1.40). The 730 nm laser beam was passed through a Chroma ZET730/10x clean-up filter prior to expansion. Lasers were switched on and off using Uniblitz V14 shutters. The focusing lens was placed on a vertical translation stage to permit manual adjustment of the TIRF angle. Emission light was separated from excitation light with a multipass dichroic mirror (ZT405/488/532/640/730rpc-uf2and; Chroma) mounted in an Olympus BX filter cube, and laser lines were further attenuated with a ZET405/488/532/640 m emission filter (Chroma) and StopLine 488/532/635 (Semrock) and ZET730nf (Chroma) notch filters. A home-built beamsplitter[39] was used to separate Cy3B emission from Cy5/Cy7 emission using a Chroma T640lpxr dichroic. Chroma ET650sp and 488/532 m emission filters were placed in the Cy3B emission path, and a filter wheel was placed in the Cy5/7 emission path

to select for Cy5 emission (ET700/75 m; Chroma) or Cy7 emission (ET811/80 m; Chroma). These two channels were imaged on separate halves of an electron-multiplying charge-coupled device camera (Hamamatsu, ImageEM 9100-13), which was operated at maximum EM gain. A motorized microstage (Mad City Labs) was used to position the sample and move between fields of view. The microscope was controlled by Hamamatsu HCImage live version 4.4.0.1 and Labview version 15.0f2.

Microfluidic chambers were constructed in the following manner: a Dremel tool with a diamond-tipped rotary bit was used to drill two holes 10 mm apart in a glass microscope slide; PE20 tubing was inserted into one hole and PE60 tubing into the other (Intramedic), and the tubing was cut flush on one side of the slide and fixed in place with epoxy (Devcon) on the other; double-sided SecureSeal Adhesive Sheet (Grace Bio-Labs), into which a 1.5 × 12 mm channel had been cut, was placed on the non-tubing side of the slide, aligning the channel with the holes in the slide. A glass coverslip, functionalized with a mixture of methoxypolyethylene glycol-succinimidyl valerate, MW 5000 (mPEG-SVA-5000; Laysan Bio, Inc.) and biotin-methoxypolyethylene glycol-succinimidyl valerate, MW 5000 (biotin-PEG-SVA-5000; Laysan Bio, Inc.) as previously described[39], was then placed on the second side of the adhesive sheet, and the edges of the coverslip were sealed with epoxy.

Single-molecule experiments were generally performed as follows, with modifications noted in the following sections. Solutions were drawn into the chamber by attaching a 1 mL syringe to the PE60 tubing. Flow cells were incubated with 1 mg/mL streptavidin (Sigma) in PBS for ~2 min. Unbound streptavidin was washed out with ELBS and biotinylated DNA substrates were incubated in the channel at a concentration yielding appropriate surface density (typically ~1 nM, diluted in ELBS). Unbound DNA was washed out with ELBS and experiments were performed as indicated below. Extracts were supplemented with pBMS6 (100 ng/μL); ATP (3 mM); phosphocreatine (15 mM); creatine phosphokinase (0.01 mg/mL; Sigma); protocatechuic acid (PCA; 5 mM); protocatechuate-3,4-dioxygenase (PCD; 0.1 μM); ascorbic acid (1 mM); and methyl viologen (1 mM). PCA/PCD constitute an oxygen scavenging system[40] and ascorbic acid/methyl viologen suppress fluorophore blinking[41].

## Two-color single-molecule assay for SR synapsis

The two-color single-molecule assay for SR synapsis (Fig. 2) was performed as previously described with slight modifications[10]. The biotinylated, Cy3B/Cy5 DNA substrate was immobilized on a glass coverslip in a microfluidic chamber as described above. Extracts were immunodepleted of XRCC4 and supplemented with SEC buffer or 50 nM unlabeled Lig4-XRCC4 variant, as well as pBMS6 (100 ng/μL); ATP (3 mM); phosphocreatine (15 mM); creatine phosphokinase (0.01 mg/mL; Sigma); protocatechuic acid (PCA; 5 mM); protocatechuate-3,4-dioxygenase (PCD; 0.1 μM); ascorbic acid (1 mM); and methyl viologen (1 mM). Extract was introduced to the chamber and images were taken continuously at a rate of 2 frame/s, alternating between one frame of 532 nm excitation and one frame of 641 nm excitation. Each field-of-view (FOV) was imaged for 2 min, and five FOVs were imaged per experiment. Surface laser power density was measured through the objective with epi-illumination using a Coherent FieldMate power meter with an OP-2 VIS detector (532 nm: 8 W/cm$^2$; 641 nm: 4 W/cm$^2$). Synapsis events were automatically detected as stepwise increases in FRET$_E$ using a custom MATLAB script. Synapsis rates were calculated as the number of synapsis events divided by the total time DNA molecules were unsynapsed with both fluorophores intact.

## Two-color single-molecule assay for Lig4 colocalization and DNA binding

The two-color single-molecule assay for Lig4 colocalization and DNA binding (Fig. 3) was performed as described in the preceding section

with the following modifications: DNA substrate was labeled with Cy3B on both ends; extracts were supplemented with 10 nM Cy5-labeled Lig4-XRCC4 variants; images were taken continuously at a rate of 20 frame/s, alternating between one frame of 532 nm excitation and one frame of 641 nm excitation; one FOV was imaged for 6 min per experiment; laser powers were 16 W/cm$^2$ (532 nm) and 16 W/cm$^2$ (641 nm). For the experiment in Supplementary Data Fig. 3c, 641 nm laser power was increased to 48 mW/cm$^2$. To ensure analyzed DNA spots had two intact Cy3B fluorophores, minimum DNA spot intensity was set at a threshold determined by the intensities of spots showing two Cy3B photobleaching events. See "Single-molecule data analysis" below for further details of colocalization and FRET analysis.

### Three-color single-molecule assay for synapsis, Lig4 colocalization/DNA binding

The three-color single-molecule assay for synapsis and Lig4 colocalization/DNA binding (Figs. 4, 5) was performed as in the two-color single-molecule assay for SR synapsis with the following modifications: extracts were supplemented with 20 nM Cy7-labeled wild-type Lig4-XRCC4; images were taken continuously at a rate of 5 frame/s with the following iterated sequence: 532 nm ex. with Cy3B/Cy5 emission, 532 nm ex. with Cy3B/Cy7 emission, 641 nm ex. with Cy5 emission, 641 nm ex. with Cy7 emission, 730 nm ex. with Cy7 emission; laser powers were 40 W/cm$^2$ (532 nm), 40 W/cm$^2$ (641 nm), and 300 W/cm$^2$ (730 nm).

To convert Cy7-Lig4 intensities to stoichiometries, stepwise changes (corresponding to Lig4 binding/unbinding events) were detected in the Cy7 ex./Cy7 em. trajectory for each DNA molecule using the MATLAB `ischange` function. Cy7 intensities were normalized by the average step intensity for each molecule to generate a Lig4 stoichiometry estimate. Synapsis events were automatically detected as stepwise increases in Cy3B/Cy5 FRET$_E$ using a custom MATLAB script.

### Single-molecule data analysis

A publicly available automated analysis pipeline[42] was used for spot detection and determination of local background-corrected fluorescence intensities and Lig4 colocalization events. To determine Lig4 colocalization events within the pipeline, thresholds were set for minimum Lig4 spot intensity and displacement from the DNA spot, and events were required to last for at least two frames with at least two frames between successive colocalization events. Thresholds were chosen to minimize non-specific colocalization events at ROIs lacking DNA signal. Fluorescence intensities were exported from the pipeline and corrected for bleed through from donor channel to acceptor channel, direct excitation of acceptor fluorophore by donor excitation laser, and differences in fluorophore quantum yield and detection efficiency (gamma factor[43]). Corrected intensities were used to calculate apparent FRET efficiencies. For two-color experiments, Cy3B/Cy5 FRET$_E$ was calculated as Cy5 intensity upon Cy3B excitation divided by the sum of Cy3B and Cy5 intensity upon Cy3B excitation ($E_{35} = I_{35}/(I_3 + I_{35})$). For three-color experiments, apparent FRET efficiencies were calculated as follows: $E_{35} = I_{35}/(I_3 + I_{35} + I_{37})$; $E_{37} = I_{37}/(I_3 + I_{35} + I_{37})$; $E_{57} = I_{57}/(I_5 + I_{57})$.

### Reporting summary

Further information on research design is available in the Nature Portfolio Reporting Summary linked to this article.

## Data availability

Source data are provided with this paper. Raw, uncropped gel images are available on Zenodo (https://doi.org/10.5281/zenodo.10055272). Additional data supporting the findings of this study are available from the authors upon request. Source data are provided with this paper.

## Code availability

Single-molecule experiment analysis was performed using a published Matlab pipeline[42], available at [https://github.com/quantitativenanoscopy/cosmos_pipeline]. Post-processing for visualization was performed using custom Matlab code, available from the authors upon reasonable request.

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

## Acknowledgements

We thank members of the Loparo and Walter laboratories for helpful discussions and comments on the manuscript. We acknowledge the support of the Center for Macromolecular Interactions at Harvard Medical School for mass photometry experiments. This work was supported by National Institutes of Health grant R01GM115487 (to J.J.L.). J.C.W. is an investigator of the Howard Hughes Medical Institute and an American Cancer Society Research Professor.

## Author contributions

B.M.S. performed all experiments and analyzed the data. S.M.C. designed and performed initial characterization of Lig4 DNA binding mutations. B.M.S, S.M.C., J.C.W. and J.J.L. conceived experiments and wrote the paper.

## Competing interests

The authors declare no competing interests.

## Ethics

The Institutional Animal Care and Use Committee of the Harvard Medical Area Standing Committee on Animals approved relevant protocols.
