## [Peer Review File · Nature Communications]

Structural role for DNA Ligase IV in promoting the fidelity of non-homologous end joiningREVIEWER COMMENTS

Reviewer #1 (Remarks to the Author):

The manuscript reports an ensemble and single-molecule analysis of the mechanism by which DNA Ligase IV plays its structural role during NHEJ in frog egg extracts. The work, which is a follow-up of PMID: 31862156, is brilliantly executed (with elegant 3-color experiments) and beautifully documented. Overall, the work brings about new insights on the adaptability and reversibility of the NHEJ reaction.

Comments :

1) About the depletion/reconstitution experiments. Extracts were treated with anti-XRCC4 to deplete extracts of XRCC4 and DNA Ligase IV assuming that XRCC4 and DNA ligase IV exist exclusively in complex. In human cells, the pool of XRCC4 is largely more abundant than the pool of DNA Ligase IV. Is this the case in frog egg extracts? In the reported experiments, XRCC4 depleted extracts were supplemented with recombinant XRCC4/DNA Ligase IV complex. How is the system responding to complementation of the extracts with "free" XRCC4 in addition to XRCC4/DNA Ligase IV?

2) No major differences in DNA Ligase IV adenylation were observed. Rates should have been measured.

3) To this reviewer, it remains unclear and somewhat confusing whether a "synaptic" LR complex assembles or not. After reading PMID: 33854234 related to the human system, it seems that the DNA Ligase IV is required.

4) Discussion: about the notion of "one strand ligation" and in support of this idea, the reviewer noticed in Figure 1C early appearance of OC species (nicked circles resulting from one-strand ligation) while SC species (resulting from two-strand ligation) appear later and are thus kinetically separable in this system.

Reviewer #2 (Remarks to the Author):

This paper by Stinson et al. very elegantly monitors the XRCC4-Ligase IV complex as it participates in DNA end joining reactions observed using single-molecule co-localization and TIRF microscopy. By imaging with three simultaneous channels the authors are able to observe the stoichiometry of Lig4 in the DNA end-repair complex and by various FRET combinations determine whether the DNA ends are held far or close apart in the long-range or short-range synaptic complexes for instance. Overall the manuscript provides insight into the comings and goings of the Lig4 in the complex and its stoichiometry during the different phases of the process.

The main concern with the manuscript is that it is insufficiently clear whether the DNA ends which have been modified with dyes for the purposes of the assay are competent for ligation and normal interaction with Lig4 or not (eg Fig. 2, Fig. 3, Fig. 4, Fig. 5), and this raises questions as one is reading the manuscript as to whether or not what the authors observe is actually on-pathway. In Fig. 2 for instance is there any evidence that the short-range complex that is formed in fact leads to repair of the DNA? That evidence could beneficially be added to supplementary materials. In Fig. 3, it is hard for the reader not to wonder: what is one to make of the repeated binding/unbinding events by Lig4 -- have the ends been repaired but the enzyme keeps revisiting the site nonetheless? Or is it not able to repair the ends and therefore keeps revisiting the site? Are these single-Lig4 events and therefore lacking the second Lig4 which the authors later imply is required for successful repair (but again without providing clear evidence that successful repair has happened). Are the same dynamics observed if the DNA ends are not phosphorylated? Although it is raised relative to Fig. 3, this question regarding phosphorylation can and should be asked at every key assay of the study as a central control which necessarily sheds light on the dynamics

observed in the presence of phosphate groups. In Fig. 4B and the other events observed in this assay, has DNA repair happened or not? The author's graphical abstract of the event suggests not (but if so why not?) but one could wonder if repair has not in fact taken place (and again that then illuminates for the reader whether what is being observed in the leadup is on-pathway or not). Thus it looks like Cy3-Cy5 FRET continues after 50s and the increase in Cy3B emission likely comes from loss of transfer to a Lig4-Cy7 rather than loss of Cy5 FRET. Thus could this in fact correspond to repair?

To resume, this is technically and experimentally a very nice piece of work of significant interest to the field, but the take-home message remains muddled and its force unrealized because of the way in which the functionality of the DNA ends and the repair reaction itself are insufficiently addressed.

Minor Point

Fig. 5, it is difficult to understand why, under Cy7 excitation, the Cy7 emission level drops at 25 seconds, but no drop is observed in Cy7 emission under either Cy3B or Cy5 excitation.

We thank the reviewers for their thoughtful critiques of our work. In response, we have performed several new experiments (new Extended Data Figs. 1c, 1d, 1e, 2a, 2b, 2e, and 2i) and modified the manuscript text (changes highlighted in yellow). We address each of the reviewer comments individually below.

Reviewer #1 (Remarks to the Author):

The manuscript reports an ensemble and single-molecule analysis of the mechanism by which DNA Ligase IV plays its structural role during NHEJ in frog egg extracts. The work, which is a follow-up of PMID: 31862156, is brilliantly executed (with elegant 3-color experiments) and beautifully documented. Overall, the work brings about new insights on the adaptability and reversibility of the NHEJ reaction.

Comments :

1) About the depletion/reconstitution experiments. Extracts were treated with anti-XRCC4 to deplete extracts of XRCC4 and DNA Ligase IV assuming that XRCC4 and DNA ligase IV exist exclusively in complex. In human cells, the pool of XRCC4 is largely more abundant than the pool of DNA Ligase IV. Is this the case in frog egg extracts? In the reported experiments, XRCC4 depleted extracts were supplemented with recombinant XRCC4/DNA Ligase IV complex. How is the system responding to complementation of the extracts with “free” XRCC4” in addition to XRCC4/DNA Ligase IV?

The reviewer is correct to point out that although Lig4 is constitutively complexed with XRCC4, the converse is not true, and there is a pool of “free” XRCC4 in human cells. XRCC4 also appears to be in modest molar excess relative to Lig4 in frog egg extracts by quantitative mass spectrometry estimates (~1.5-fold excess of XRCC4; Wühr et al., *Current Biology* (2014)). We performed an additional experiment to test the effect of supplementing extracts with free XRCC4 following XRCC4 immunodepletion (new Extended Data Figure 1e; text lines 116-118). As we observed previously (Graham et al., *Molecular Cell* (2016)), free XRCC4 alone does not rescue end joining (lanes 13-16). Moreover, free XRCC4 does not enhance rescue of end joining by Lig4/XRCC4 (compare lanes 9-12 and 17-20). We conclude that the pool of free XRCC4 has little functional significance during the NHEJ reaction in our system. Excess XRCC4 may help ensure efficient Lig4/XRCC4 complex formation, which is required for Lig4 stability (Bryans et al., *Mutation Research* (1998)).

2) No major differences in DNA Ligase IV adenylation were observed. Rates should have been measured.

We have repeated the Lig4 adenylation experiments in Extended Data Figure 1c and measured relative adenylation rates. These experiments revealed modest adenylation rate defects for our DNA-binding mutants (see new Extended Data Figure 1c; text line 107). It is possible that the loss of positive charge associated with

these mutations affects the efficiency of ATP binding to Lig4, given the long-range nature of electrostatic interactions.

More broadly, the purpose of this experiment is to demonstrate that severe defects in short-range synapsis we observe with these mutants are not due to gross defects in protein folding. We note that the defects in adenylation rates are much more modest than the defects in short-range synapsis rates (Figure 2c). In addition, all Lig4 DNA-binding mutants co-purified with XRCC4 (Extended Data Figure 1a), suggesting the protein is not globally misfolded. Finally, we performed new mass photometry experiments to test for the presence of aggregates in our protein preparations. These experiments identified Lig4-X4 complexes of the expected mass with little aggregation (new Extended Data Figure 1D; text lines 109-110).

3) To this reviewer, it remains unclear and somewhat confusing whether a “synaptic“ LR complex assembles or not. After reading [PMID: 33854234](https://pubmed.ncbi.nlm.nih.gov/33854234/) related to the human system, it seems that the DNA Ligase IV is required.

We thank the reviewer for raising the lack of clarity on this point. Our prior work in egg extracts demonstrated that the long-range synaptic complex formation does not require Lig4 and always precedes formation of the short-range synaptic complex. We believe this result is fully consistent with the recent cryo-EM studies of NHEJ synapsis, which have observed two structurally distinct LR complexes. In the first LR complex structure, DNA end synapsis is mediated entirely by DNA-PKcs and Ku in the absence of Lig4-XRCC4 (Chaplin et al., *Nature Structural & Molecular Biology* (2020)). In the second LR complex structure (Chen et al., *Nature* (2021); cited by the reviewer), two Lig4-XRCC4 complexes interact with a single XLF homodimer to help mediate synapsis. In our view, the latter structure is likely an intermediate between the Lig4-independent LR synaptic complex and the Lig4-dependent SR synaptic complex. We have modified Figure 6 to more clearly reference the LR synaptic complex, and we have modified the text to clarify this point (see lines 286-289).

4) Discussion: about the notion of “one strand ligation” and in support of this idea, the reviewer noticed in Figure 1C early appearance of OC species (nicked circles resulting from one-strand ligation) while SC species (resulting from two-strand ligation) appear later and are thus kinetically separable in this system.

The reviewer makes an astute observation regarding the OC and SC species in Figure 1c. However, the OC species may be a mixture of nicked plasmids, as the reviewer suggests, and fully repaired plasmids that have not yet been sufficiently chromatinized to run as SC products. Because of this confounding factor, we are unable to use the reviewer’s observation as support for the “one strand ligation” model.

Reviewer #2 (Remarks to the Author):

This paper by Stinson et al. very elegantly monitors the XRCC4-Ligase IV complex as it participates in DNA end joining reactions observed using single-molecule co-localization and TIRF microscopy. By imaging with three simultaneous the authors are able to observe the stoichiometry of Lig4 in the DNA end-repair complex and by various FRET combinations determine whether the DNA ends are held far or close apart in the long-range or short-range synaptic complexes for instance. Overall the manuscript provides insight into the comings and goings of the Lig4 in the complex and its stoichiometry during the different phases of the process.

The main concern with the manuscript is that it is insufficiently clear whether the DNA ends which have been modified with dyes for the purposes of the assay are competent for ligation and normal interaction with Lig4 or not (eg Fig. 2, Fig. 3, Fig. 4, Fig. 5), and this raises questions as one is reading the manuscript as to whether or not what the authors observe is actually on-pathway. In Fig. 2 for instant is there any evidence that the short-range complex that is formed in fact leads to repair of the DNA? That evidence could beneficially be added to supplementary materials.

We thank the reviewer for raising this important point. Our previous work has demonstrated that fluorophores near DNA ends have no significant effect on end joining (Graham et al., Molecular Cell (2016)). We now demonstrate this in the current manuscript in two new figure panels: new Extended Data Figure 2a, an ensemble assay showing the presence of fluorophores has no significant effect on end joining kinetics (text lines 134-135); and new Extended Data Figure 2b, a single-molecule assay as in Figure 2, which demonstrates that the vast majority of high-FRET molecules persist as such following an SDS wash, indicating that the DNA has been ligated (text lines 135-138).

We have also examined whether fluorophores near the DNA ends affect Lig4 interaction with DNA. When the fluorophore is moved far from the DNA ends, Lig4 colocalization kinetics are essentially unaltered (new Extended Data Figure 2e; text lines 164-166).

From these experiments and our prior results, we conclude that our observations with fluorophore-modified DNA are on-pathway, fluorophores near DNA ends do not substantially alter the Lig4 interaction with DNA, and short-range synapsis, as measured by our smFRET assays, very frequently results in DNA ligation.

In Fig. 3, it is hard for the reader not to wonder: what is one to make of the repeated binding/unbinding events by Lig4 -- have the ends been repaired but the enzyme keeps revisiting the site nonetheless? Or is it not able to repair the ends and therefore keeps revisiting the site? Are these single-Lig4 events and therefore lacking the second Lig4 which the authors later imply is required for successful repair (but again without providing clear evidence that successful repair has happened).

We thank the reviewer for noting the lack of clarity in our explanation of the data presented in Figure 3. The repeated binding/unbinding events the reviewer

mentions are essentially unaltered under conditions where short-range synapsis and repair are blocked (Extended Data Figure 2f). Thus, these dynamics represent transient DNA binding events that do not result in successful repair in the vast majority of cases. This observation underscores the importance of our three-color assay (Figs. 4 and 5) which directly measures the relatively rare Lig4-DNA binding events that result in short-range synapsis and ligation. We have modified the text to clarify this point (see lines 185-186).

Are the same dynamics observed if the DNA ends are not phosphorylated? Although it is raised relative to Fig. 3, this question regarding phosphorylation can and should be asked at every key assay of the study as a central control which necessarily sheds light on the dynamics observed in the presence of phosphate groups.

We thank the reviewer for pointing out that DNA end chemistry may alter observed Lig4 dynamics. We have repeated the experiment in Figure 3 with unphosphorylated DNA ends and observe no significant changes in Lig4 dynamics (new Extended Data Figure 2i; text lines 195-197). Thus, the presence of a terminal phosphate appears to have little effect on these transient DNA binding events. This result is consistent with previous EMSA experiments in which lack of a 5' phosphate had little effect on Lig4-DNA binding (Reid et al., *Nucleic Acids Res.* (2017)). Our results do not exclude that DNA end chemistry plays an important role in productive synapsis, as suggested by Reid et al.

We believe that further investigation of DNA end chemistry in synapsis and modulating dynamics of end-modifying factors deserves comprehensive treatment in an independent study. We share the reviewer's interest in comparisons between 5' phosphate and 5' hydroxyl ends; we believe it would be similarly interesting to examine how other DNA ends (e.g., overhangs of varying length and degree of microhomology) affect synapsis and interactions with Lig4. Moreover, as we point out in the Discussion, we believe that the detailed interplay between Lig4 and processing enzymes on incompatible DNA ends is a critical unanswered question. We hope to address these questions in a subsequent study.

In Fig. 4B and the other events observed in this assay, has DNA repair happened or not? The author's graphical abstract of the event suggests not (but if so why not?) but one could wonder if repair has not in fact taken place (and again that then illuminates for the reader whether what is being observed in the leadup is on-pathway or not). Thus it looks like Cy3-Cy5 FRET continues after 50s and the increase in Cy3B emission likely comes from loss of transfer to a Lig4-Cy7 rather than loss of Cy5 FRET. Thus could this in fact correspond to repair?

In our three-color assay, repair will result in FRET between Cy3B and Cy5. In Figure 4B, we do not observe FRET between Cy3B and Cy5, and repair has not occurred. The reviewer correctly notes that the increase in Cy3B emission at ~50s is due to loss of energy transfer to Cy7-Lig4 upon its dissociation. Figure 5 shows persistent FRET between Cy3B and Cy5, starting around 25 s, consistent with DNA repair. Although we lack a real-time single-molecule measurement of ligation, and

therefore cannot state with certainty when ligation occurs for a given molecule, our results in new Extended Data Figure 2b demonstrate that molecules with high FRET between Cy3B and Cy5 are very often ligated. We have modified the text to clarify the ligation status of DNA ends in our single-molecule experiments (lines 136-138).

To resume, this is technically and experimentally a very nice piece of work of significant interest to the field, but the take-home message remains muddled and its force unrealized because of the way in which the functionality of the DNA ends and the repair reaction itself are insufficiently addressed.

We thank the reviewer for the opportunity to develop the take-home message of our work more fully. We believe our new experiments showing that fluorescent labels on DNA ends do not affect ligation or interaction with Lig4 substantially clarify the transactions between Lig4 and DNA ends.

Minor Point

Fig. 5, it is difficult to understand why, under Cy7 excitation, the Cy7 emission level drops at 25 seconds, but no drop is observed in Cy7 emission under either Cy3B or Cy5 excitation.

We thank the reviewer for pointing out the lack of clarity on this point. Within the time resolution of our experiment, the departure of one Cy7-Lig4 (which results in the drop in Cy7 emission) is virtually simultaneous with short range synapsis (in which the remaining Cy7-Lig4 binds both DNA ends, such that FRET from both Cy3B and Cy5 to Cy7 is maintained). We expect that higher time resolution would allow observation of a transient drop in either Cy3B-Cy7 or Cy5-Cy7 FRET, representing the state in which one Cy7-Lig4 has dissociated but the remaining Cy7-Lig4 has yet to bind the newly released DNA end. Unfortunately, technical limitations of our assay do not allow for this time resolution. We have modified the text to clarify this point (see lines 255-256).

REVIEWERS' COMMENTS

Reviewer #1 (Remarks to the Author):

The authors have adequately addressed the concerns of the reviewers.

Reviewer #2 (Remarks to the Author):

The authors have done an outstanding job clarifying the points raised, and this reviewer strongly recommends its publication.